Allele specific expression analysis identifies regulatory variation associated with stress-related genes in the Mexican highland maize landrace Palomero Toluqueño

Aguilar-Rangel M. Rocío 1 2
Chávez Montes Ricardo A. 1 3
González-Segovia Eric 1
Ross-Ibarra Jeffrey 4
Simpson June K. 2
Sawers Ruairidh J.H. rusawers@cinvestav.mx ruairidh.sawers@gmail.com 1
1 Unidad de Genómica Avanzada (LANGEBIO), Centro de Investigación y de Estudios Avanzados del Instituto Politécnico Nacional (CINVESTAV-IPN) , Irapuato , Guanajuato , Mexico
2 Departamento de Ingeniería Genética, Centro de Investigación y de Estudios Avanzados del Instituto Politécnico Nacional (CINVESTAV-IPN) , Irapuato , Guanajuato , Mexico
3 ABACUS: Laboratorio de Matemáticas Aplicadas y Cómputo de Alto Rendimiento del Departamento de Matemáticas, Centro de Investigación y de Estudios Avanzados del Instituto Politécnico Nacional (CINVESTAV-IPN) , Ocoyoacac , Estado de México , Mexico
4 Department of Plant Sciences, Center for Population Biology and Genome Center, University of California , Davis , CA , United States of America
VanBuren Robert
Electronic publication date: 2017 Aug 23
Publication date: 2017
Volume: 5
Electronic Location ID: e3737
Received 2017 Jun 22; Accepted 2017 Aug 4
Copyright: ©2017 Aguilar-Rangel et al.
Copyright year: 2017
Copyright holder: Aguilar-Rangel et al.
License: This is an open access article distributed under the terms of the Creative Commons Attribution License, which permits unrestricted use, distribution, reproduction and adaptation in any medium and for any purpose provided that it is properly attributed. For attribution, the original author(s), title, publication source (PeerJ) and either DOI or URL of the article must be cited.
License URL: https://creativecommons.org/licenses/by/4.0/

Keywords: Abiotic stress, Allele specific expression, Maize, Palomero Toluqueño

Funding: UC-MEXUS CN-15-1476 National Science Foundation No. 1546719 Consejo Nacional de Ciencia y Tecnología EDOMEX-2011-C01-165873 National Science Foundation IOS-1027527 This work was supported by UC-MEXUS (CN-15-1476), the National Science Foundation (No.1546719) and the Consejo Nacional de Ciencia y Tecnología (EDOMEX-2011-C01-165873 to RACM). Unpublished RNA-Seq data shared by Patrick Schnable (Iowa State University) was generated using support from the National Science Foundation (IOS-1027527). The funders had no role in study design, data collection and analysis, decision to publish, or preparation of the manuscript.

==============================
Background

Gene regulatory variation has been proposed to play an important role in the adaptation of plants to environmental stress. In the central highlands of Mexico, farmer selection has generated a unique group of maize landraces adapted to the challenges of the highland niche. In this study, gene expression in Mexican highland maize and a reference maize breeding line were compared to identify evidence of regulatory variation in stress-related genes. It was hypothesised that local adaptation in Mexican highland maize would be associated with a transcriptional signature observable even under benign conditions.

Methods

Allele specific expression analysis was performed using the seedling-leaf transcriptome of an F1 individual generated from the cross between the highland adapted Mexican landrace Palomero Toluqueño and the reference line B73, grown under benign conditions. Results were compared with a published dataset describing the transcriptional response of B73 seedlings to cold, heat, salt and UV treatments.

Results

A total of 2,386 genes were identified to show allele specific expression. Of these, 277 showed an expression difference between Palomero Toluqueño and B73 alleles under benign conditions that anticipated the response of B73 cold, heat, salt and/or UV treatments, and, as such, were considered to display a prior stress response. Prior stress response candidates included genes associated with plant hormone signaling and a number of transcription factors. Construction of a gene co-expression network revealed further signaling and stress-related genes to be among the potential targets of the transcription factors candidates.

Discussion

Prior activation of responses may represent the best strategy when stresses are severe but predictable. Expression differences observed here between Palomero Toluqueño and B73 alleles indicate the presence of cis-acting regulatory variation linked to stress-related genes in Palomero Toluqueño. Considered alongside gene annotation and population data, allele specific expression analysis of plants grown under benign conditions provides an attractive strategy to identify functional variation potentially linked to local adaptation.

Introduction

Extensive study across different plant species has identified a range of transcriptional responses to abiotic stresses. Although basic responses are typically conserved, variation in the regulation of stress-responsive genes has been observed among individuals and varieties, potentially playing an important role in adaptation to stressful environments (Hannah et al., 2006; Swanson-Wagner et al., 2012; Rengel et al., 2012; Lasky et al., 2014). From an agronomic perspective, biotechnological approaches to enhance crop stress tolerance to abiotic stress often aim to manipulate gene expression rather than engineer protein sequences (e.g., Kamthan et al., 2016). Similarly, efforts to identify suitable material for breeding towards these same goals have drawn on natural cis-acting regulatory variation acting on stress-responsive gene expression (e.g., Mao et al., 2015). As such efforts are intensified in the face of mounting concern regarding the impact of climate change on crop productivity, there is ever greater interest in the genetic basis of variation in stress-responses (Des Marais, Hernandez & Juenger, 2013).

Crop landrace varieties represent an invaluable genetic resource. Collectively, the range of environments exploited by landraces typically exceeds that of improved varieties, and individual landraces may be adapted to conditions that would be considered stressful in conventional agriculture (Ruiz Corral et al., 2008; Romero Navarro et al., 2017). Nonetheless, although landraces represent a compelling source for enhancing abiotic stress tolerance in breeding programs, the task of identifying useful genetic variants and transferring them to breeding material is far from trivial (Sood et al., 2014). In addition to the complication of working with often heterogenous landrace germplasm, reproducing stress conditions for evaluation is costly and difficult. Furthermore, stress is not well reflected by a single experimental treatment, but rather represents a continuous environmental range defined by interacting variables acting over the lifetime of the plant. Large-scale phenomics efforts are an attempt to implement the factorial designs required to capture such complexity (Houle, Govindaraju & Omholt, 2010; Furbank & Tester, 2011), but they require a substantial investment in infrastructure that may not be feasible in many research contexts. A number of approaches aim to leave aside such difficulties, and to identify candidate genes directly from genomic data through the incorporation of environmental variables into population genetic and genome wide association studies (Coop et al., 2010; Lasky et al., 2015). Here, as a further alternative, transcriptome data is explored for signatures of an enhanced stress response hardwired in locally adapted material, and evident under benign conditions.

Stress responses are considered to be an adaptation to an unpredictable, often suboptimal environment. Under benign conditions, however, activation of these same pathways, by exogenous application of plant hormones or mutation of genes involved in signaling pathways, is associated with growth retardation (Staswick, Su & Howell, 1992; Hu et al., 1996; Bowling et al., 1997; Ellis & Turner, 2001), indicating both their potential cost to the plant, and the benefit of maintaining tight regulation. Nonetheless, when conditions are adverse, but predictably so, it may be advantageous to anticipate activation of stress pathways and avoid the delay between stimulus and response inherent in plasticity (Levins, 1968; Von Heckel, Stephan & Hutter, 2016). In cultivated systems, non-adapted varieties can benefit from mild priming stress treatments that activate protective mechanisms and prepare the plants for future more severe environmental challenges (Van Hulten et al., 2006; Hilker et al., 2016). In practice, however, the first exposure to a stress may be severe, placing the unprepared organism at risk. Here, the hypothesis is addressed that anticipation of stress responses is a hallmark of local adaptation in marginal environments, presenting an opportunity to identify genetic variation related to enhanced stress tolerance that is expressed under benign conditions.

Comparative transcriptome analysis of stress tolerant and non-tolerant varieties provides a powerful approach to identify the molecular mechanisms underlying tolerance variation (e.g., Hayano-Kanashiro et al., 2009; Von Heckel, Stephan & Hutter, 2016). The number of differentially accumulating transcripts, however, may be large, and the data reflect both cis-acting and trans-acting regulatory variation. Critically, per se comparison of varieties has little power to characterize the genetic architecture of stress tolerance or to identify causative genetic variation. In addition, when material is diverse, phenological differences can make it difficult to devise an appropriate sampling strategy. With the development of sequencing based methods to study the transcriptome, it is possible to make use of natural sequence variation to quantify allele specific expression (ASE) in F1 hybrid individuals generated from the cross of two different lines of interest (Springer & Stupar, 2007b; Springer & Stupar, 2007a; Zhang & Borevitz, 2009; Lemmon et al., 2014). Characterization of ASE in F1 material avoids the problems of comparing parents that may be very different in growth and development by evaluating both alleles within the same cellular environment, directly revealing cis-acting genetic variation for transcript accumulation (Springer & Stupar, 2007b; Lemmon et al., 2014; Waters et al., 2017).

In this study, a transcriptome dataset was examined for evidence of cis-regulatory variation linked to stress-associated genes in Palomero Toluqueño (PT), a maize landrace originating from the highlands of Central Mexico (Prasanna, 2012; Perales & Golicher, 2014). The Mexican highland environment exposes maize plants to a number of abiotic stresses: bringing plants to maturity under low-temperatures necessitates planting early in the year, exposing seedlings to late frosts and water deficit before onset of the annual rains; throughout the growing season, low-temperature, high-levels of UV radiation and hail storms pose further challenges (Eagles & Lothrop, 1994; Lafitte & Edmeades, 1997; Jiang et al., 1999; Mercer, Martínez-Vásquez & Perales, 2008; Ruiz Corral et al., 2008). The unique group of Mexican highland maize landraces, including PT, has been shown previously to be superior in the highland niche to maize originating from temperate, mid-altitude tropical or lowland tropical regions, with respect to seedling emergence, photosynthetic efficiency, and tolerance to frost, cold, drought and hail (Eagles & Lothrop, 1994; Mercer, Martínez-Vásquez & Perales, 2008). To identify evidence of regulatory variation that might underlie adaptation to these conditions, an F1 was generated between PT and the midwest-adapted maize reference line B73, and the leaf transcriptome analyzed under benign greenhouse conditions to detect ASE. Results of the analysis were compared with a published study in which B73 seedlings were exposed to cold, heat, salt and UV stress treatments (Makarevitch et al., 2015). A total of 277 genes were identified showing a pattern of ASE under benign conditions that mirrored the response of the same gene under stress in B73, hereafter referred to as prior stress response (PSR). The PSR candidate set included transcription factors and genes associated with plant hormone signalling, a number of which are discussed in more detail and presented as candidates for future functional analysis.

Materials and Methods

Plant material, RNA preparation, and sequencing

Seed of the Mexican highland landrace Palomero Toluqueño accession Mexi5 was obtained from the International Maize and Wheat Improvement Center (CIMMYT; stock GID 244857). The original collection was made near the city of Toluca, in Mexico state (19.286184N, −99.570871W), at an elevation of 2,597 masl. An F1 hybrid stock was generated from the cross between the inbred line B73 and PT, grown under standard greenhouse conditions (27 °C day, 24 °C nights; 15 h days, 9 h nights; 30% humidity. Ames, Iowa) and total RNA was extracted from leaf tissue of a single, 14 day-old seedling using the Qiagen RNeasy Plant Mini Kit (cat ID 74904) according to the manufacturer’s protocol. RNA integrity was assessed by spectrophotometry and agarose gel electrophoresis. Library preparation was performed using the Illumina protocol as outlined in the TruSeq RNA Sample Preparation Guide (15008136 A, November 2010) and paired-end sequencing was carried out on the Illumina HiSeq 2000 platform. Raw data is available in the NCBI (http://www.ncbi.nlm.nih.gov) Sequence Read Archive under accession SRP011579. Plant growth conditions in the Makarevitch study (Makarevitch et al., 2015) were described previously in the published report. Briefly, whole above ground tissue was collected for 14 day old seedlings. For cold stress, seedlings were incubated at 5 °C for 16 h. For heat stress, seedlings were incubated at 50 °C for 4 h. For high salt stress, plants were watered with 300 mM NaCl 20 h prior to tissue collection. For UV stress, plants were irradiated using UV-B lamps for 2 h prior to tissue collection.

Allele Specific Expression (ASE) analysis

Allele specific expression (ASE) analysis was based on the method of Lemmon and collaborators (Lemmon et al., 2014) and the detailed pipeline is presented as Data  S1 (pipeline). A set of 39475 B73 transcripts was generated by selecting the longest predicted transcript for each gene annotated in the AGPv3.22 B73 reference genome (ftp://ftp.ensemblgenomes.org/pub/release-22/). Six transcripts whose sequences consisted of only, or mostly, undefined (N) bases were removed (GRMZM2G031216_T01, GRMZM2G179334_T01, GRMZM2G307432_T01, GRMZM2G316264_T01, GRMZM2G406088_T01 and GRMZM2G700875_T01), resulting in a set of 39,469 sequences. A total of 151,168,196 paired-end reads from the B73xPT F1 transcriptome were trimmed using Trimmomatic (Bolger, Lohse & Usadel, 2014) and aligned using bwa mem (Li, 2013) to the set of B73 transcripts. The resulting alignment was processed using samtools, bcftools and vcfutils (Li et al., 2009; Li, 2011a; Li, 2011b) to identify polymorphisms. We then created a set of PT pseudo-transcripts by substituting the identified sequence variants into the B73 reference transcripts. A single fasta file was created that contained two sequences per locus, one B73 transcript and one PT pseudo-transcript, and B73xPT F1 reads were re-aligned to this F1 pseudo-reference using bowtie2 (Langmead & Salzberg, 2012) with eXpress (Roberts et al., 2011; Roberts & Pachter, 2013) recommended parameters. The number of reads per B73 and/or PT transcript was then quantified using eXpress. A total of 9,256 transcripts were identified to contain polymorphisms, allowing estimation of ASE. Genes were considered to show ASE when the number of associated reads assigned to B73 or PT transcripts was significantly different (χ2 test against an equal number of counts; p < 0.05; Bonferroni correction for multiple tests) and the absolute log2-transformed ratio of PT/B73 reads was >1.

Gene Ontology annotation, enrichment analyses and comparison of ASE genes to published data

Candidate ASE genes were assigned to Gene Ontology categories (release 52 available at ftp://ftp.gramene.org/pub/gramene). Obsolete annotations were replaced by the corresponding “consider” or “replaced_by” category(ies) in the ontology file (go.obo) available at http://www.geneontology.org/ (dated 2016-09-19). Categories associated with at least 10 genes were considered in further analysis. Enrichment analyses were performed comparing ASE candidates against the 9,256 polymorphic gene set, using the Bingo (Maere, Heymans & Kuiper, 2005) Cytoscape (Shannon et al., 2003) plugin, using a hypergeometric test and controlling for multiple tests using Benjamini and Hochberg False Discovery Rate at 1%. Categories were called PT up or PT down using a threshold of abs(median logFC) ≥1. Enrichment analysis of the 1938 TF targets gene set was performed against the 11,299 genes represented on the microarray (see “Reconstruction of gene co-expression network” section below).

Candidate ASE genes were cross-referenced to a published study describing transcriptional responses in maize seedlings exposed to cold, heat, salt and UV stresses (Makarevitch et al., 2015). Although a number of inbred lines were analyzed in the Makarevitch study, only the B73 data was used in the comparison with the B73xPT transcriptome. Genes were considered to show prior stress response (PSR) with respect to a given stress when: (1) identified as ASE; (2) responding significantly to stress in the Makarevitch study (absolute log2 fold change >1; called as significant in the Makarevitch study; calls “up” or “on” in the published study were considered here as “up”, similarly, “down” or “off” were considered as “down”); (3) the sign of ASE was concordant with the sign of stress response.

Fst values for population level differentiation between Mesoamerican and South American highland and lowland maize populations (Takuno et al., 2015) were obtained from https://github.com/rossibarra/hilo_paper/tree/master/fst; where multiple SNPs were associated with a single gene, the values reported correspond to the SNP showing the highest Fst in Mesoamerica.

Reconstruction of gene co-expression network

Publicly available maize Affymetrix microarray data was downloaded from the ArrayExpress website (http://www.ebi.ac.uk/arrayexpress/; experiments E-GEOD-10023, E-GEOD-12770, E-GEOD-12892, E-GEOD-18846, E-GEOD-19785, E-GEOD-22479, E-GEOD-28479, E-GEOD-31188, E-GEOD-40052, E-GEOD-41956, E-GEOD-48406, E-GEOD-48536, E-GEOD-54310, E-GEOD-59533, E-GEOD-69659, E-MEXP-1222, E-MEXP-1464, E-MEXP-1465, E-MEXP-2364, E-MEXP-2366, E-MEXP-2367, E-MEXP-3992). Low quality CEL files identified using the arrayQualityMetrics (Kauffmann, Gentleman & Huber, 2009) R package were discarded. Using the sample data relationship file (sdrf) associated with each experiment, samples for B73 leaves were selected, resulting in a high quality, homogeneous dataset of 165 CEL files.

Probeset sequences for the maize Affymetrix microarray were aligned using seqmap (Jiang & Wong, 2008) to the AGPv3.22 transcripts with no mismatches allowed, and probesets whose probe sequences did not align or aligned to transcripts corresponding to more than one locus were discarded. Probesets that were represented by less than 4 probe sequences were also discarded. This resulted in a list of 11,299 probesets that unambiguously matched one locus. The list of 11,299 probesets was used to create a custom chip definition file (CDF) using the ArrayInitiative python package (http://wellerlab.uncc.edu/ArrayInitiative/), and to filter the original Affymetrix Maize.probe_tab file to create a custom probe_tab file. The custom CDF and custom probe_tab file were then used to create the corresponding cdf and probe_tab R packages using the makecdfenv (Irizarry et al., 2006) and AnnotationForge (Carlson & Pages, 2017) R packages, respectively. The microarray name in the 165 CEL files was then modified to match the custom cdf and probe_tab packages name, and these modified CEL files were normalized using gcrma (Wu & Gentry, 2017). The resulting normalized dataset was then used as input for the ARACNE algorithm (Margolin et al., 2006a; Margolin et al., 2006b), and inference was carried out for the 7 ASE and stress-responsive transcription factors (see ‘Results’) at DPI 0.1 as previously described (Chávez Montes et al., 2014).

Results

A total of 2,386 genes exhibited allele specific expression in the B73xPT F1 hybrid

To identify regulatory variation associated with stress-related genes, high throughput sequencing was used to quantify transcript abundance in leaves harvested from an F1 seedling generated from the cross between the Mexican highland landrace PT and the reference line B73. Alignment to the B73 reference gene models identified 9,256 genes containing at least one sequence variant that could be used to distinguish the products of B73 and PT alleles. For 2,386 (26%) of these 9,256 polymorphic transcripts, the number of reads corresponding to the B73 allele differed significantly (p < 0.05; Bonferroni correction for multiple tests) from the number of reads corresponding to the PT allele with an absolute log2 fold change >1, and these genes were considered to exhibit allele specific expression (ASE; Data S2 [F1_ counts]). For 1,412 (59%) of the ASE candidate genes, accumulation of the PT transcript was lower than that of the B73 transcript (log2 PT/B73 <−1; hereafter, “PT-down”), while for the remaining 974 (41%) of the ASE candidates, the PT transcript was accumulated at higher levels (log2 PT/B73 >1; hereafter, “PT-up”).

To obtain an overview of the ASE candidates, a Gene Ontology (GO) analysis was performed. The set of 2386 ASE candidates was not enriched for any specific GO categories with respect to the 9,256 polymorphic gene set, but, nonetheless, many individual genes belonged to biological processes categories related to stress responses, including responses to heat (GO: 0009408), cold (GO: 0009409) and salt (GO: 0009651) (Fig. 1). Overall, 52 biological process categories were represented by at least 10 genes. Of these, 38 (73%) were PT-down (based on the median log2 PT/B73 of the associated genes), and 11 (21%) were PT-up, and the remaining three categories had a median log2 PT/B73 close to 0 (Data S3 [ASE_loci_GO_P]). A similar pattern was observed for molecular function categories: 57 categories were associated with at least 10 ASE genes, 42 PT-down, 12 PT-up and three showing no trend (Data S4 [ASE_loci_GO_F]).

Figure 1 ASE candidate genes are assigned to a range of biological process Gene Ontology categories.

Hierarchical tree of Gene Ontology biological process categories represented in ASE loci. Nodes represent categories, with the root GO:0008150 biological process as the uppermost node. Edges represent the parent-child (i.e., “is_a”) relationship between categories. Node color indicates the median ASE (log2 PT/B73) for the genes in the category, with light blue indicating negative values and dark red indicating positive values. Node size is proportional to the number of loci assigned to corresponding category. Some category names were abbreviated for clarity.

A total of 277 genes showed prior stress responses

To identify evidence of prior stress response (PSR) in PT, the ASE gene set was compared with a previous study reporting changes in the transcriptome of B73 seedlings exposed to cold, heat, salt or UV treatments (Makarevitch et al., 2015). Of these treatments, cold and UV stress are directly relevant to plant performance in the highland niche, and salt stress may be considered to some extent a proxy for drought conditions. PT is not predicted to be adapted to heat stress, and, as such, the heat treatment provides an interesting contrast to the other conditions, although, as described below, many genes in this study were responsive to multiple stresses. A total of 1,407 stress responsive genes identified in the Makarevitch study were present also in the 9,256 polymorphic gene set for which ASE had been evaluated (Data S2 [F1_counts]). Of these 1,407 genes, 432 (31%) showed ASE, a slight enrichment compared with the 2,386 (26%) ASE genes in the 9,256 polymorphic gene set as a whole (ASE, Makarevitch: 432; ASE, non-Makarevitch: 1,963; non-ASE, Makarevitch: 984; non-ASE, non-Makarevitch: 5,886; χ2 = 15.7, d.f. = 1, p < 0.001). From this 432 gene set, a gene was considered to exhibit PSR in PT if the sign of ASE was concordant with the sign of B73 stress response: i.e., PT-up and induced by stress in B73, or PT-down and repressed by stress in B73. On this basis, a set of 277 PSR candidates was identified (Figs. 2A–2D; Data S5 [Maka_can_ annot]). The majority of these 277 genes respond to two or more stress treatments (Figs. 3A–3C), but often in different directions such that they present stress-specific PSR (Figs. 3B–3C): 194 were identified as showing PSR with respect to one treatment, 62 with respect to two, 17 with respect to three, and 4 with respect to all four (Fig. 3C). Of the 277 genes, 92 showed PSR with respect to cold, 65 with respect to heat, 136 with respect to salt, and 92 with respect to UV (Fig. 3B). The number of PSR genes with respect to any given stress was proportional to the number of genes responding to that stress in the 1,407 polymorphic gene set (cold, PSR: 92, non-PSR: 631; heat, PSR: 65, non-PSR: 374; salt, PSR: 136, non-PSR: 736; UV, PSR: 92, non-PSR: 444; χ2 = 5.2, d.f. = 3, p = 0.16), and there was no indication of an enrichment for PSR with respect to any one of the four treatments. In contrast to the complete ASE gene set, the majority of the 277 PSR genes were PT-up (181 PT-up, 96 PT-down; Data S5 [Maka_can_annot]), although this general trend was not observed when the UV treatment was considered alone, where the majority of PSR genes were PT-down (34 PT-up, 58 PT-down; Fig. 2D).

Figure 2 ASE identifies PSR in PT with respect to B73.

ASE (log2 PT/B73) in control F1 leaves for the 1,407 sequence variant, stress-responsive gene set against B73 stress response (log2 stress/control) for (A) cold, (B) heat, (C) salt and (D) UV treatments as reported in the Makarevitch dataset. Numbers in each quadrant represent the count of genes called as significant in ASE and stress comparisons. In each plot, the quadrants represent (clockwise from upper left) genes up ASE/down stress, up ASE/up stress, down ASE/up stress, down ASE/down stress. Genes called as up ASE/up stress or down ASE/down stress are considered to show PSR and are shown as filled circles. Other genes are shown as points. Axes through the origin are shown as red dashed lines. A small number of genes outside the axis range are not shown, but are considered in the gene count.

Figure 3 PSR candidates may respond to multiple stresses in B73.

(A) Number of genes from the 277 PSR gene set that responded to cold, heat, salt, UV or a combination of stresses in the Makarevitch B73 study. (B) Number of genes called as PSR in PT with respect to each stress from the same 277 gene set. (C) Counts with respect to number of stresses of genes in A and B. Numbers above bars give counts.

Hormone related genes and transcription factors showed constitutive stress responses in PT

A primary aim of the analysis was the definition of a small number of candidate genes for future functional analysis. For this purpose, the PSR candidate genes were cross-referenced with the classical maize gene list, a curated set of 4,908 well-annotated genes, many linked with existing functional data (the “combined set” gene list was obtained from www.maizegdb.org/gene_center/gene and filtered for unique gene identifiers). Of the 277 PSR candidate genes, 48 were present in the classical gene list (Fig. 4; Data S5 [Maka_ can_annot]), including 9 genes associated with hormone homeostasis (Table 1) and 12 transcription factors (TFs; Table 2; Jin et al., 2017) that were considered of special interest. The 277 PSR candidates were cross referenced with a published study of population level differentiation between Mesoamerican and South American highland and lowland maize (Takuno et al., 2015). Fst estimates and significance were reported for 183 of the PSR candidates, 22 which showed significant differentiation (p < 0.1) between highland and lowland Mesoamerican populations, including the hormone associated gene Czog1 (GRMZM2G168474; Data S5 [Maka_can_annot]). The number of PSR candidates showing significant Fst was as expected based on the overlap with the 1,407 polymorphic gene set (Fst reported for 1,032 of 1,407 genes; PSR, Fst p < 0.1: 22; PSR, Fst p >  = 0.1: 161; non-PSR, Fst p < 0.1: 100; non-PSR, Fst p >  = 0.1: 749; χ2 <  1, d.f. = 1, p = 1). To gain insight into potential TF targets and their role in stress responses, a gene co-expression network for the PSR TFs was generated using available maize Affymetrix microarray data and the ARACNE algorithm. Seven of the 12 TFs were unambiguously identified in the maize Affymetrix microarray probeset, and were co-expressed with 1,938 genes (Data S6 [tfs_ASE_01_suppl]). Co-expressed genes represent potential targets of TF action, and, as such, may not themselves exhibit ASE. Indeed, of the 1,938 genes associated with the 7 TFs, 1,097 were present in the polymorphic gene set, but only 239 showed ASE. A total of 344 of the 1,938 co-expressed genes (17%) were responsive to one or more stress treatments in the Makarevitch dataset (Fig. 5). A GO analysis detected enrichment in the 1,938 gene co-expression set with respect to translation, photosynthesis and non-mevalonate isoprenoid pathway categories (Data S7 [Bingo_aracne]).

Figure 4 Classical PSR candidate genes.

Heatmap representation of ASE (log2 PT/B73) and B73 response to cold, heat, salt and UV stress (log2 stress/control) as reported in the Makarevitch dataset for PSR candidates in the maize classical gene list. Asterisks (*) in the stress columns indicate a given gene was called as PSR with respect to that stress.

Table 1 ASE and stress-responsive hormone-related genes.

List of genes involved in hormone biosynthesis, transport or catabolism present in the 277 PSR gene set. ASE call indicates biased expression of the PT allele (1) or B73 allele (−1). Response to stress indicates the name of the stress for which the gene was called as differentially expressed in the Makarevitch dataset. Prior stress response indicates the stress condition for which the sign of the ASE call and the stress response coincide.

Gene id	Symbol	Molecular function	Hormone	ASE call	Response to stress	Prior stress response	
GRMZM2G070563	–	auxin efflux carrier	auxin transport	1	heat, salt, uv	heat, salt	
GRMZM2G072632	–	auxin efflux carrier	auxin transport	1	heat, salt, uv	heat, salt	
GRMZM2G112598	–	auxin efflux carrier	auxin transport	1	heat, salt, uv	heat, salt	
GRMZM2G475148	–	auxin efflux carrier	auxin transport	1	heat, salt	heat, salt	
GRMZM2G072529	Acco31	1-aminocyclopropane- 1-carboxylate oxidase	ethylene biosynthesis	1	cold, heat, salt, uv	cold, heat, salt, uv	
GRMZM2G020761	–	putative cytochrome P450 (castasterone C-26 hydroxylase)	brassinosteroid catabolism	−1	cold, salt, uv	cold, uv	
GRMZM2G148281	Opr7	12-oxo-phytodienoic acid reductase	jasmonate biosynthesis	−1	salt, uv	salt	
GRMZM2G168474	Czog1	cis-zeatin O-glucosyl transferase	cytokinin homeostasis	1	salt	salt	

Table 2 ASE and stress-responsive TFs.

List of TFs present in the 277 PSR gene set. PlantTFDB family indicates the TF family according to the PlantTFDB (Jin et al., 2017). ASE call indicates biased expression of the PT allele (1) or B73 allele (−1). Response to stress indicates the name of the stress for which the gene was called as differentially expressed in the Makarevitch dataset. PSR indicates the stress condition for which the sign of the ASE call and the stress response coincide. In Affymetrix array indicates whether the TF is represented in the maize Affymetrix microarray.

Gene id	Symbol	PlantTFDB family	ASE call	Response to stress	Prior stress response	In affymetrix array?	
GRMZM2G159937	Bhlh57	bHLH	1	cold, salt, uv	cold, uv	no	
GRMZM2G148333	Ereb202	ERF	1	uv	uv	yes	
GRMZM2G010920	Glk18	G2-like	−1	heat, uv	uv	no	
GRMZM2G127537	Hb11	HD-ZIP	1	salt, uv	salt	yes	
GRMZM2G041127	Hb54/ZmHdz10	HD-ZIP	1	cold, heat, salt	cold	yes	
GRMZM2G049695	Mybr24	MYB-related	1	salt, uv	salt, uv	no	
GRMZM2G121753	Mybr89	MYB-related	−1	cold, salt, uv	uv	no	
GRMZM2G127379	NacTF25/ZmNAC111	NAC	−1	cold, salt, uv	cold	no	
GRMZM2G162739	NacTF5	NAC	−1	cold, salt, uv	salt	yes	
GRMZM2G003715	NacTF61	NAC	1	cold, uv	cold, uv	yes	
GRMZM2G312201	NacTF70	NAC	1	uv	uv	yes	
GRMZM2G071907	Wrky50	WRKY	1	salt	salt	yes	

Figure 5 Co-expression networks for PSR TFs and their putative stress-responsive targets.

Nodes represent genes and edges represent co-expression as calculated by the ARACNE algorithm at DPI 0.1. (A) Network of seven PSR TFs (labeled centres of circles) with their co-expressed, stress-responsive (genes called up/on or down/off in the Makarevitch dataset) putative targets. Triangles indicate genes that were called as presenting ASE. (B–E) Network filtered to retain only co-expressed genes responsive to (B) cold, (C) heat, (D) salt or (E) UV treatments, as indicated. In the filtered networks the red and blue colors indicate up or down regulation (as log2 FC from the Makarevitch dataset), respectively, under the corresponding stress.

Discussion

From a starting set of 9,256 polymorphic genes, we identified 2,386 genes presenting allele specific expression (ASE) in seedling leaves of a B73xPT F1 hybrid individual. Comparison of the ASE gene list with a published dataset reporting B73 stress responses (Makarevitch et al., 2015) identified a subset of 277 (out of 432) prior stress response (PSR) candidate genes exhibiting a bias in transcript accumulation between PT and B73 alleles that mirrored the B73 response to one or more stress treatments. No enrichment was observed in GO term assignments in either the ASE gene set or the PSR gene set. Nonetheless, given that ASE is assaying cis-acting variation, a small number of genes associated with a given GO term may have biological significance. The ASE gene set showed a bias towards lower expression of the PT allele, reflected in the observation that the median value of ASE for the majority of GO categories associated with ASE genes was also negative. Contrary to this trend, the subset of 277 selected PSR candidates showed a bias towards higher expression of the PT allele (181 of 277 presented higher expression of the PT allele), also reflected in the 1,407 polymorphic genes that overlapped with the Makarevitch set.

The bulk of the PSR gene set (206 of 277) responded to two or more stresses in the Makarevitch B73 data, although in the majority (194 of 277) of cases the PSR itself was with respect to a single stress only (Fig. 3), indicating that in many cases the sign (up/down) of the response in B73 differed between stresses (Data S5 [Maka_ can_annot]). By definition, a gene could not show PSR with respect to both of two different stresses if the B73 responses were opposing. There was no evidence that genes showing opposing stress responses in B73 were less likely to show ASE, and consequently, PSR in PT—indeed, such genes were actually better represented in the 277 PSR gene set (156 of 277; 56%) than in the 1,407 polymorphic and stress-responsive gene set (511 of 1,407; 36%). As such, many ASE events may appear contradictory with respect to any given stress, i.e., PT-up ASE in genes repressed by B73 under stress, or PT-down ASE in genes induced by B73, especially in the context of cold and UV treatments, against which PT is considered to be well adapted. The spatio-temporal dynamics of stress responses, however, are complex (e.g., Secco et al., 2013), and the resolution of the present analysis, based on single time points and tissues, is limited. For example, the previously characterized salt associated HD-ZIP transcription factor Hb54 (also named ZmHdz10, GRMZM2G041127; Zhao et al., 2011; Zhao et al., 2014) showed PT-up ASE, but was repressed by salt treatment in the Makarevitch dataset, and consequently not considered to show PSR. In this case, however, an additional functional study reports Hb54 to indeed be induced by salt treatment (Zhao et al., 2014), albeit at a different time point, and with a different treatment than that applied in the Makarevitch study (300 mM NaCl for 20 h in Makarevitch et al.; 200 mM NaCl for 3–12 h in Zhao et al.). The study of Zhao and colleagues reports also that constitutive expression of Hb54 in Arabidopsis and rice increases ABA sensitivity and tolerance to drought and salt stress. In light of these data, PT-up ASE of Hb54 may indeed have biological relevance, reflected by the number and nature of associated co-expression candidates (Fig. 5). In the absence of further characterization, it would be premature to discount the potential phenotypic impact, or adaptive value, of other examples where ASE in PT is opposed to the B73 stress response reported in the Makarevitch data.

Previous studies have highlighted the importance of cis-acting regulatory variation in driving diversity in plant stress responses (e.g., Waters et al., 2017). The generation of novel physiological strategies to confront stress conditions may be most efficient when a change in the regulation of a single gene has multiple, coordinated downstream consequences. Mechanistically, two functional categories of clear interest are hormones, systemic regulators of physiology at the whole plant level, and transcription factors (TFs), with their capacity to impact multiple downstream targets through a regulatory cascade. The 277 PSR gene list includes eight hormone-related genes (Table 1), including genes implicated in the metabolism of cytokinin (Czog1; Martin et al., 2001), jasmonate (ZmOpr7; Yan et al., 2012) and ethylene (Acco31; Gallie & Young, 2004; Avila et al., 2016). Additional PSR candidates included Ks2 (GRMZM2G093526; ZmKSL5), a gene related to the ent-kaurene synthase required for gibberellin biosynthesis, but more likely involved in the more specialized kauralexin A series biosynthesis pathway (Fu et al., 2016), and Thi2 (GRMZM2G074097), encoding a thiamine thiazole synthase activity required for synthesis of the thiazole moiety during the production of thiamin (vitamin B1; Woodward et al., 2010). With regard to the latter candidate, B vitamins, although not strictly plant hormones, can play an analogous role in whole plant physiology in the face of stress (Hanson et al., 2016). Thiamin application has been reported to alleviate the impact of abiotic stress in a number of crops, including maize (e.g., Kaya et al., 2015), and thiamin synthesis has been proposed as a target for transgenic biofortification (e.g., Dong, Stockwell & Goyer, 2015). Identification of PT-up ASE associated with Thi2 represents a compelling target for further analysis. Furthermore, both Thi2 and the related gene Thi1 (GRMZM2G018375) were also co-expressed with the PT-up ASE drought and salt associated HD-ZIP TF Hb54 (Zhao et al., 2014; Table 2; Fig. 5; Data S6). Interestingly, the PSR candidates Czog1 and Ks2 were reported previously to show significant population level differentiation between highland and lowland mesoamerican maize populations (Fst; p = 0.004, p = 0.04, respectively; Takuno et al., 2015), indicating that variation at these loci may indeed play a role in local adaptation.

In total, twelve TFs were present in the 277 PSR candidate gene set (Table 1), including four NAC TFs. The NAC TFs are a plant-specific family implicated broadly in abiotic stress responses (Nakashima et al., 2012; Puranik et al., 2012; Nuruzzaman, Sharoni & Kikuchi, 2013; Nakashima, Yamaguchi-Shinozaki & Shinozaki, 2014), previously proposed as a target for engineering multiple stress tolerance (Shao, Wang & Tang, 2015). The potential role of NAC TFs in a generalized stress response is reflected by the observation that the candidates ZmNacTF5 (GRMZM2G162739), ZmNacTF25 (also named ZmNac111, GRMZM2G127379; Mao et al., 2015), ZmNacTF61 (GRMZM2G003715) and ZmNacTF70 (GRMZM2G312201) responded to three, three, two and one stress treatments, respectively (Table 2). The genes ZmNacTF5 and ZmNacTF25 showed PT-down ASE, and PSR with respect to salt and cold, respectively, while the genes ZmNacTF61 and ZmNacTF70 showed PT-up ASE and PSR with respect to cold and UV, respectively. In B73, insertion of a miniature inverted-repeat transposable element (MITE) in the ZmNacTF25 promoter has been reported previously to be associated with reduced gene expression (relative to a number of tropical lines) and increased susceptibility to drought (Mao et al., 2015). The accumulation of ZmNacTF25 transcripts in B73, however, is reduced under cold in the Makarevitch dataset, indicating a potential trade-off between temperate and tropical lines, and possible relevance of the PT-down ASE in the highland niche. The gene ZmNacTF61 was notable for strong PT-up ASE (log2 PT/B73 =3.26 and 2.15), up-regulation under both cold and UV stress, and association with a large number (116) of strongly cold- and UV- induced co-expression candidates, including the jasmonate biosynthetic genes Opr7 and Lox4 (Figs. 4 and 5; Data S6).

Candidate PSR genes presented here were identified on the basis of ASE under benign conditions. Investigation of the degree to which ASE is maintained under stress conditions is required to determine whether the level of expression of these candidates remains plastic in PT, albeit with an expression level different from B73, or whether expression has been canalized to a constitutively responsive state (Waddington, 1942; Levins, 1968; Von Heckel, Stephan & Hutter, 2016). Nonetheless, the potential to identify relevant cis-regulatory variation through exploration of the transcriptome under benign conditions presents an attractive avenue to investigate stress response and local adaptation. A number of the candidates identified here suggest testable predictions regarding hormone accumulation and expression of candidate TF targets in the PT landrace. In a number of cases, ASE was observed in genes reported previously to show significant genetic differentiation between lowland and highland Mexican maize populations, offering further evidence of a link to adaptation to the highland niche (Data S5 [Maka_can_ annot]). A recent study in monkey flower (Mimulus guttatus) using ASE analysis to compare locally adapted coastal and inland accessions has found cis-regulatory effects to be the main driver for regulatory variation, providing a precedent for the approach proposed here (Gould, Chen & Lowry, 2017). Validation of specific candidate genes will require functional characterization, but it is anticipated that this will be greatly facilitated by continued development of resources for maize reverse genetics and the generation of introgression lines derived from Mexican highland maize.

Conclusions

Expression differences were observed between PT and B73 alleles under benign conditions that mirror the B73 response to cold, heat, salt and/or UV treatments. The observed patterns of expression indicate the presence of cis-acting regulatory variation differentiating the PT landrace from the B73 reference inbred. Regulatory variants linked to classical genes associated with signaling and stress-responses potentially contribute to the adaptation of PT to the Mexican highland environment.

Supplemental Information

Supplemental Information 1 Analytical pipeline

Click here for additional data file.

Supplemental Information S2 39,469 AGPv3.22 longest transcripts and PT/B73 F1 transcript counts

Click here for additional data file.

Supplemental Information S3 GO biological process categories associated with the 2386 ASE genes

Click here for additional data file.

Supplemental Information S4 GO molecular function categories associated with the 2386 ASE genes

Click here for additional data file.

Supplemental Information S5 277 ASE and stress-responsive genes

Click here for additional data file.

Supplemental Information S6 Annotated co-expression data set for B73 leaves and the 7 TFs represented in the maize Affymetrix microarray

Click here for additional data file.

Supplemental Information S7 GO biological process Bingo enrichment analysis for the 1,938 co-expressed gene set

Click here for additional data file.

We acknowledge Patrice Dubois for assistance in the generation of F1 seed stock, and Patrick Schnable and Cheng Ting Yeh for generation of transcriptome data.

Additional Information and Declarations

Competing Interests

Author Contributions

DNA Deposition

Data Availability

Jeffrey Ross-Ibarra is an Academic Editor for PeerJ.

M. Rocío Aguilar-Rangel analyzed the data, wrote the paper, prepared figures and/or tables, reviewed drafts of the paper.

Ricardo A. Chávez Montes wrote the paper, prepared figures and/or tables, reviewed drafts of the paper.

Eric González-Segovia and Jeffrey Ross-Ibarra wrote the paper, reviewed drafts of the paper.

June K. Simpson reviewed drafts of the paper.

Ruairidh J.H. Sawers conceived and designed the experiments, performed the experiments, contributed reagents/materials/analysis tools, prepared figures and/or tables, wrote the paper, reviewed drafts of the paper.

The following information was supplied regarding the deposition of DNA sequences:

Transcriptome data is available from the NCBI (http://www.ncbi.nlm.nih.gov) Sequence Read Archive under accession number SRP011579.

The following information was supplied regarding data availability:

Raw data is in the form of transcriptome data available in from the NCBI Sequence Read Archive.

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
