# Peer review of "Allele specific expression analysis identifies regulatory variation associated with stress-related genes in the Mexican highland maize landrace Palomero Toluqueño"

_PeerJ, doi:10.7717/peerj.3737_

## Round 0.1 · original submission · Major Revisions

Your manuscript entitled “Allele specific expression analysis identifies regulatory variation associated with stress-related genes in the Mexican highland maize landrace Palomero Toluqueño” has been seen by two qualified reviewers. Based on their detailed assessment and my own, I feel this manuscript would be well suited for publication in PeerJ following a detailed revision. In their revision, the authors should address each of the issues raised by both reviewers. While I agree with reviewer 2 that replication of the RNAseq data as well as multiple independent F1s would be ideal, these additional experiments are not required.

·

Basic reporting

The quality of the writing is clear and professional throughout.

In the introduction, it would help the readership a lot of introduce some of the background/theory on the cost of constitutive activation of stress responsive genes. The authors describe why this can sometimes be a good thing when the stress is repeated and predictable in most growing seasons, but that point makes a lot more sense if presented in the context that constitutive induction of stress responsive genes general produces a decrease in growth rate and/or yield when grown under ideal conditions. It might also be useful to introduce some of the papers that attempt to use the adaptation of landraces to particular environments as a trait in GWAS studies, as this would seem to support the general approach taken by the authors here. Laskey 2015 doi: 10.1126/sciadv.1400218 is the example I am most familiar with in that genre.

Figures and tables are clear and easy to read. I believe figure 5 could be omitted, but if the authors like it, it obviously doesn't have to go.

Results are relevant to the hypotheses presented in the introduction.

Experimental design

The experimental design is clearly stated and this does in fact constitute original research.

Several small additional pieces of information should be provided as part of the methods section:

Lines 205-210: What was the statistical test or theshold value used to distinguish between a biological process category which would be classified as either PT up or PT down rather than close to zero?

Lines 238-239: Can you cite a specific source or criteria used for your set of 4908 well annotated genes? Going to the link provided, I found three lists: "Classical Genes" N=437,"MaizeGDB Curated Genes" N=4868 and "Combined" N=5303.

Lines 254-256: Need to define what background gene set was used in this GO analysis (presumably not the 9,256 polymorphic genes used above).

Validity of the findings

The data presented is robust and I generally agree with the statistical tests employed by the authors. I have several points however that may need further consideration though:

Lines 231-232: Could you quantify the bias present in the 1407 gene set too for comparison here? (Apologies if the degree of bias was introduced elsewhere and I missed it).

Lines 242-246: Is the overlap between the CR candidates and the FST dataset more, less or equal to the amount of overlap expected between these datasets?

Lines 272-275: This seems strikingly high. If 206 genes respond to two or more stresses, let's take the simplest case where it is only two stresses. If the direction of response to stress is random and equal proportions of genes are up and downregulated, one quarter of these genes should be upregulated in both stresses, and one quarter downregulated in both species for a total of 50%. So we'd expect at least 103 genes which were CR in at least two stresses. If there is a bias towards up or down regulation in response to stress, that number would increase. If some 206 genes that respond to two or more stresses are actually differentially expressed in response to three or more stresses (Fig 3) that number would increase further.

Lines 278-281: I am not sure the statistics are correct here. If a gene shows opposite sign changes in different B73 stresses, than 100% of ASE genes will be classified as CR genes. If a gene only shows one direction of change in B73 in response to stress then only 50% of ASE genes should be classified as CR genes.

Lines 142-143: Not a criticism, but I was very happy to see that the authors used the correct background population set (the 9,256 polymorphic genes) in their enrichment analysis. I was worried they were going use the set of all annotated genes as many people do.

Additional comments

The authors describe the concept of constitutive gene activation as a form of canalization. Generally I hear the term canalization used in the context of compensatory changes that allow a trait to remain constant across a range of conditions that would otherwise produce variation in the trait. Gene expression is indeed a molecular trait, but it isn't clear to me that the default behavior of a gene in the absence of specific regulation to the contrary wouldn't be to remain expressed at a constant level across multiple environments. Particularly since, as the authors point out on lines 345-348, this dataset doesn't actually show that PT alleles has lost its ability to be up or down regulated in response to stress, only that it does so from a higher baseline expression level, I would urge the authors to either reconsider the use of the term canalization in this context, or, if they still feel the term is appropriate, more explicitly label these discussions as speculation in the absence of allele specific expression level data from B73/PT hybrids under stressed conditions.

Lines 89-91: Is PT adapted to better tolerate hail than non-highlands adapted landraces?

Lines 212-233: Was there any enrichment for constitutive stress response in PT relative to the opposite pattern (so a gene that is upregulated in response to stress in B73, but the PT allele is expressed to a lower level than the B73 allele in the F1 hybrid? or down in response to stress in B73, but the PT allele is expressed at higher levels in the F1). Given that the putatively constitutively stress responsive genes are the primary focus of the paper, it would help the interpretation of the results to know and state if this pattern is more common than expected of randomly up or downregulated alleles in PT.

Reviewer 2 ·

Basic reporting

No comments. The text is clearly written, and the figures are publication quality.

Experimental design

While the experimental setup is interesting, several aspects of the study need to be addressed before it can be considered for publication. First, the conclusions are based on analysis of a single hybrid individual, with no replication. It cannot be concluded from the authors’ dataset that the results represent a signal of adaptation, or are merely noise in the expression data. Additional biological and/or technical replicates of the hybrid, and control genotypes, would make the conclusions more robust to noise.

Second, the Materials and Methods section lacks sufficient detail to properly interpret and reproduce the results of the study. While the authors do a good job at documenting their sequence analysis and bioinformatics workflows, they do not adequately describe how their plant material was grown. The authors state that they reared the F1 hybrid individual under benign conditions, but do not state specifically what the greenhouse conditions were. They also do not compare their growing conditions to those used in the Makarevitch study, which makes it difficult to compare their observed expression response to those reported in the previous study. Additional detail in the Materials and Methods about the benign greenhouse conditions, and how they compare to the Makarevitch stress conditions, would make interpretation of the observed expression responses clearer.

Validity of the findings

See above section for comments regarding reproducibility.

What is the interpretation for a gene that shows expression bias toward the Palomero Toluqueño allele, but was implicated in heat or salt stress by the Makarevitch study? It is not clear what can be concluded from this result, since the authors state that cold and UV are the primary stresses in the highland environments where Palomero Toluqueño is grown.

Additional comments

In addition to the larger concerns above, I have the following minor points:

Line 59: Missing year in Levins citation.
Line 105: “Near to the city” should be “near the city”
Lines 107-108: Was RNA extracted from just the leaves of the seedling, or the entire seedling?
Lines 124-125: Why was BWA-MEM chosen for read alignment, as opposed to a splice-aware read mapping tool such as HISAT2 or STAR?
Lines 224, 321: What test is being done to assess statistical significance of FST?

---

## Round 0.2 · accepted · Accept

Your revised manuscript has addressed the previous concerns held by both reviewers and I feel it is now suitable for publication in PeerJ.